Data gap or biodiversity gap? Evaluating apparent spatial biases in community science observations of Odonata in the east-central United States

Bullion Christian M.
Bahlai Christie A. cbahlai@kent.edu
Department of Biological Sciences, Kent State University , Kent , OH , United States of America
Huber Dezene
Electronic publication date: 2024 Sep 30
Publication date: 2024
Volume: 12
Electronic Location ID: e18115
Received 2023 Feb 15; Accepted 2024 Aug 27
Copyright: ©2024 Bullion and Bahlai
Copyright year: 2024
Copyright holder: Bullion and Bahlai
License: This is an open access article distributed under the terms of the Creative Commons Attribution License, which permits unrestricted use, distribution, reproduction and adaptation in any medium and for any purpose provided that it is properly attributed. For attribution, the original author(s), title, publication source (PeerJ) and either DOI or URL of the article must be cited.
License URL: https://creativecommons.org/licenses/by/4.0/

Keywords: Community science, Ecological modeling, Geographic bias, Insect biodiversity, Odonatology, Spatial gaps

Funding: National Science Foundation grant DBI 2045721 Graduate Student Senate Research Fellowship award from Kent State University Support for this research was provided by National Science Foundation grant DBI 2045721 to Christie A. Bahlai. Christian M. Bullion was supported by a Graduate Student Senate Research Fellowship award from Kent State University. The funders had no role in study design, data collection and analysis, decision to publish, or preparation of the manuscript.

==============================
Odonates (dragonflies and damselflies) have become popular study organisms for insect-based climate studies, due to the taxon’s strong sensitivity to environmental conditions, and an enthusiastic following by community scientists due to their charismatic appearance and size. Where formal records of this taxon can be limited, public efforts have provided nearly 1,500,000 open-sourced odonate records through online databases, making real-time spatio-temporal monitoring more feasible. While these databases can be extensive, concerns regarding these public endeavors have arisen from a variety of sources: records may be biased by human factors (ex: density, technological access) which may cause erroneous interpretations. Indeed, records of odonates in the east-central US documented in the popular database iNaturalist bear striking patterns corresponding to political boundaries and other human activities. We conducted a ‘ground-truthing’ study using a structured sampling method to examine these patterns in an area where community science reports indicated variable abundance, richness, and diversity which appeared to be linked to observation biases. Our observations were largely consistent with patterns recorded by community scientists, suggesting these databases were indeed capturing representative biological trends and raising further questions about environmental drivers in the observed data gaps.

Introduction

Community science initiatives have been crucial for understanding changes in biodiversity, distribution, and phenology, due to their potential to generate large volumes of data and cover broad geographical areas (Fraisl et al., 2022). The possible benefits of community science (often referred to as citizen science) have been well-documented and could represent a viable alternative to data acquisition for projects where scarce financial or logistical resources prevent traditional, multi-visit sampling (Lauret et al., 2021). Community science data may represent a considerable boon to academic biodiversity science as a data source for species where formal data collection is rare or incomplete, such as taxa not considered to have economic importance. For example, odonates (dragonflies and damselflies) have fostered a large and widespread hobbyist following that has provided nearly 1,500,000 open-sourced odonate records worldwide through public databases like Odonata Central and iNaturalist (https://www.inaturalist.org/)—entries that could conceivably lay the groundwork for numerous insect ecological studies. The phenology and spatial distribution of the odonates is tied closely to environmental cues and conditions, primarily temperature and photoperiod, even leading to them being referred to as ‘living barometers’ (Hassall, 2015). However, climate-driven disruptions to the delicate ‘when’ and ‘where’ of odonates could have severe implications for these insects (Zarnetske, Skelly & Urban, 2012). For example, range-shifting (as a species’ population shifts to areas with more favorable environments) and changes in phenology (the timing of life-history events) have both been well-documented climate-linked responses in odonates (Hassall & Thompson, 2008; Hassall & Thompson, 2010; Hickling et al., 2005; May et al., 2017; Winder & Schindler, 2004). Community science records gathered from platforms like iNaturalist have already been used, with great success, in the long-term monitoring of Californian odonates (Rapacciuolo et al., 2017) and to identify breeding occurrences in Oklahoma odonates (Patten et al., 2019).

However, understanding these fundamental biodiversity trends requires data that is unbiased in time and space, and community science has not gone uncriticized in these regards (Catlin-Groves, 2012; Lukyanenko, Parsons & Wiersma, 2016). Most concerningly, community science initiatives risk reflecting biases and even potentially interests held by their participants, leading to temporal, spatial and even taxonomic biases. Biases arising from infrastructure and human population density are of particular note. For example, in comparison to their observed biological richness, agricultural areas were vastly oversampled (Geldmann et al., 2016). Similarly, volunteer sampling can often be affected by a ‘cottage effect’, being more likely to sample easily accessible locations, such as those near roads or population centers (Millar, Hazell & Melles, 2019). These patterns of interaction can have profound implications: research involving bird species have suggested that biases in community science data may produce less accurate models for habitats with distinct environmental characteristics and low reporting rates (Johnston et al., 2020). In their most extreme, these low-density areas may form complete data gaps, which interfere with our understanding of assemblages and species distributions, especially in areas that are highly vulnerable to diversity loss and highly understudied (Archer et al., 2014).

Indeed, our research group encountered a striking example of what appeared to be a geographical bias in reporting frequency for community science records when initiating a study utilizing community science records to examine recent changes in odonate communities in the east-central United States (Fig. 1). In a prominent show of extreme sampling, the state of Ohio showed high-density reports across nearly its entire region, with state borders clearly visible on observation density heat maps. This pattern is driven, at least in part, by the popular Ohio Odonate Survey, a large-scale community science initiative that first ran from 1991 to 2001 and then was re-initiated in 2017. In total, this initiative has reported over 125,000 odonate observations to iNaturalist (iNaturalist, 2021; The Ohio State University, 2021). In striking contrast, a large area of the central Appalachian region of the United States to the direct south, centered around West Virginia, is seemingly extremely underrepresented in reports for, not only odonates, but a variety of taxa in community science databases (iNaturalist, 2021). This region is characterized by high elevation, temperate forests, and low human population densities, but has also experienced lower household incomes and higher poverty rates than surrounding regions in recent years (U.S. Census Bureau, 2022). While this region neighbors the National Radio Quiet Zone (NRQZ), a federally designated region of West Virginia in which radio transmissions and cellular signals are heavily restricted, potentially leading to interruptions in iNaturalist app use in this region, this NRQZ does not appear to contribute to the observed data gap, potentially due to a heightened focus on scientific research within that area.

Figure 1 iNaturalist observations of Odonata from the eastern United States, observed before January 1, 2020.

Grid cells are given with orange shading to indicate reporting density: more saturated orange indicates areas with highest reporting density. Surveyed sites are designated with black diamonds. Cities at the ends of the sampling transect are marked with grey circles, as well as Washington DC to provide spatial context. Image source code: https://github.com/cbullion/odonata-gap/blob/main/odonata_map_fig1.R. Data source: https://www.inaturalist.org/observations/export for all Odonata reports prior to Jan 1, 2020, with a latitude range of 32.90 to 43.60 and a longitudinal range of −87.36 to −74.51. There is an intermediate data file available in the repository at https://github.com/cbullion/odonata-gap/blob/main/observations-453205-1.csv.

We hypothesize that the observed data gap in central Appalachia is an artifact of human data collection patterns. A data gap like the one we observe in the iNaturalist observations in this area may arise from inherent challenges facing observation-based community science initiatives in prominently rural areas. Essentially, we predict that odonate biodiversity and abundance in central Appalachia are being under-reported by community scientists compared to areas with higher populations and access to more economic resources. If this is the case, these data collection artifacts could be shaping currently accepted species distributions through the strength of observation efforts that characterize them. Thus, we predict that between-site community trends will differ by human population density in the unstructured data produced by community scientists but will be more equally distributed across sites in structured surveys. Therefore, in this study, we set out to evaluate the reliability of unstructured surveys in documenting odonate diversity relative to structured expert sampling. As such, this study documents a ‘ground truthing’ effort where structured sampling was paired with unstructured community science records on a north-south transect, to examine any discrepancies in abundance, diversity, and community composition produced by the two data collection methods. Portions of this text were previously published as part of a preprint (https://doi.org/10.1101/2022.11.29.518107).

Materials & Methods

Study area

To evaluate patterns observed in community science records for this system, we conducted a structured survey based on comparing systematic ‘expert’ observations with publicly reported community science observation data from two major odonate open data sources, iNaturalist and Odonata Central. A north-south transect starting in the Greak Lakes basin in the northern potion of the American state of Ohio through the central Appalachians and centering on the observed data gap, provided the foundation for our structured sampling dataset and incorporated longitudinal and elevation aspects. The entire site was situated in the temperate northern deciduous ecozone.

Sampling was completed in five counties along this transect- Cuyahoga County (Ohio), Guernsey County (Ohio), Wayne County (West Virginia), Knott County (Kentucky), and Wise County (Virginia)-chosen to be approximately equally spaced, with bodies of water in naturalized areas that were reasonably accessible from roads or campgrounds (Fig. 1; Table 1). These human-accessible sites were selected as those which would reasonably represent areas where community scientist reporting would most likely occur for these counties and had physical attributes associated with ‘good’ habitat for many odonate species (open bodies of water with vegetated shores).

Table 1 Geographic information and iNaturalist reporting statistics for the five sampling locations selected for field-truthing, arranged from north to south.

The iNaturalist Users column refers to the total number of unique users providing data matching our criteria within that county.

County code	County name	Latitude (°N)	Avg. elevation (m)	iNaturalist users	iNaturalist observations	County size (km 2 )	Population density (/km 2 )	
A	Cuyahoga (OH)	41.4	174	148	1132	3230	392	
B	Guernsey (OH)	40.1	311	11	68	1370	28	
C	Wayne (WV)	38.2	296	0	0	1330	29	
D	Knott (KY)	37.3	382	3	4	910	16	
E	Wise (VA)	37.0	687	11	60	405	89	

The transect was sampled once monthly during June, July, and August, representing the peak odonate flight season for this region during 2019. For each site, data were collected from three vantage points along the lake shoreline deemed reasonably accessible to hobbyists by foot. For each vantage point at a site, surveying was conducted for ten consecutive minutes, once during the peak of the day (11:00–12:00 EST) and again during the evening of the same day (17:00 –18:00 EST), to increase the likelihood of surveying both diurnal and crepuscular populations.

Odonate sampling

For each sampling period, the number of odonates visible from the selected vantage point, facing towards the lake, was counted per species. Identification of individuals was primarily done in-hand via netting, using the Dragonflies and Damselflies of the East field guide for reference (Paulson, 2011), while identification of visually distinct species was done through observation only. Sampling was conducted by a team of two personnel, where one person made observations at all sites, and the second person served as the recorder, for consistency.

Community science data was represented by a combined Global Biodiversity Information Facility dataset, containing reports originating from iNaturalist and Odonata Central, and omitting museum records (GBIF.org, 2021). The dataset consisted of Odonate abundance data during the peak adult flight season (June, July, August) for the years 2014–2021 in the focal region. Data were subsetted to create ‘observation units’ corresponding to our structured surveys: records were aggregated by county of record and month of capture. Because variable reporting in some areas created zero-biased data incompatible with community analysis at a fine temporal scale, we combined data from multiple years to represent a ‘typical’ community that could be observed in a given place, at a given time of year.

Quantification and statistical analysis

All statistical analyses and plotting were conducted using R software (R Core Team, 2021), using the following packages: rgbif (Chamberlain & Boettiger, 2017), plyr (Wickham, 2011), ggplot2 (Wickham, 2016), vegan (Oksanen et al., 2022), MASS (Venables & Ripley, 2002), broom (Robinson, Hayes & Couch, 2022), and BiodiversityR (Kindt & Coe, 2005).

Community observation data for the focal counties was obtained from iNaturalist and Odonata Central via GBIF alongside the observations collected from the structured samples detailed above. Subsets of these datasets were created to include species identity, county, and date for each observation. For convenience and ease of understanding, specific county names were omitted and replaced with letter coding, labeling the sampled counties as A, B, C, D, and E, in order from north to south (Fig. 1, Table 1). For each dataset, count data were aggregated using the plyr package to form counts of each species per county per source. Before merging, each dataset was amended to include its source, differentiating between community observations and structured sampling observations.

To evaluate biodiversity patterns observed by method and sampling location, we computed total abundance, richness and Shannon diversity for each sample was calculated using the vegan package from the row sums of the merged dataset, with corresponding plots generated from the results. Further aggregation of the data allowed for comparisons of species counts by county per source per month. We aggregated data by month of collection to help account for varied phenology among the odonate species observed. Generalized linear models (GLM) for were constructed for abundance and species richness (with a negative binomial error structure) and diversity (using a Gaussian error structure) with sampling method, county and month as predictors constructed using the MASS package. We built models for each response variable with an interaction between sampling methods and county and one without. Model selection, through comparison of model AIC scores (a lower AIC score indicating a model has less unexplained variation and thus a better fit), was then used to evaluate the relationship of abundance, species richness and diversity between sampling method and location; a model with a better fit when the interaction was included suggests that there is geographical variation in the in the tested metric between observational methods. In contrast, a better-fitting model without an interaction effect would suggest that the same patterns are held between sites, implying that the sampling method did not vary in its ability to capture biodiversity patterns over space. This process was then repeated to evaluate odonate abundance from the same data set, with all associated plots generated using the ggplot2 package. We constructed individual-based species accumulation curves using the specaccum function in vegan, set to 100 permutations and Jacknife2 estimates for each of the sampling methods and fit a nonlinear model (method = “lomolino”) to the curves produced to estimate predicted richness by sampling method.

To examine differences in community composition among sites, sampling periods, and sampling approaches, non-metric multidimensional scaling (NMDS) using Bray-Curtis dissimilarity metrics was conducted using the vegan package. Lastly, analyses of similarities (ANOSIM) and Permutational ANOVA (adonis) was also conducted using the vegan package to determine if community composition varied between sites and sampling methodologies.

Results

This study included 1,573 observations, with 381 originating from structured sampling efforts and 1,192 originating from community science efforts. The blue dasher, Pachydiplax longipennis, was the most commonly recorded species of the 27 observed during structured efforts (13.9% of observations), while the ebony jewelwing (Calopteryx maculata) was the most commonly recorded species of the 82 observed via community science sources (7.9% of observations). Differences in abundance of records between counties was largely driven by a very high rate of reports in unstructured surveys in the northernmost county (Fig. 2). Model selection strongly favored the inclusion of an interaction term (AIC = 262.1 without interaction, = 251.5 with interaction) suggesting that the sampling methods were non-uniform across counties in their reports of number of odonates observed.

Figure 2 Boxplot comparing observed total abundance of odonates across five focal counties.

Counties are arranged north (A) to south (E), in the northeastern United States for structured and unstructured survey methods. Unstructured samples are extracted from records contributed to the Global Biodiversity Information Facility database from iNaturalist and Odonata Central community science contributions and represent all odonates reported to GBIF these sources for 2014–2012 in the months June, July and August structured surveys were completed by a trained individual conducting timed observations at a site within that county during three sampling visits (during June, July and August) in the summer of 2019. Note that there were zero observations in county ‘C’ in the unstructured surveys .

Combined observed odonate species richness was highly variable between sites, ranging from 11 to 70, with higher reported diversity at the extreme ends of the transect (Fig. 3). Community sampling efforts reported the highest richness, at 82 species, compared to the 27 species reported through structured sampling. Richness varied dramatically by county in the unstructured data, but less so in the structured surveys (Fig. 3). As with abundance, model selection strongly favored the inclusion of an interaction term (AIC = 200.7 without interaction, = 186.6 with interaction). The two northernmost counties captured the highest richness of odonates by unstructured sampling, but structured sampling captured a greater richness of odonates in the three southernmost counties. Species richness accumulated more quickly and was estimated to be dramatically higher across individuals pooled using unstructured sampling (Fig. 4). For unstructured sampling, the species accumulation curve was predicted to reach an asymptote at 226 species, where the structured sampling curve was predicted to reach an asymptote near 29.6.

Figure 3 Boxplot comparing observed species richness of odonates across five focal counties in the northeastern United States for structured and unstructured survey methods.

Data were obtained as described in Fig. 2.

Figure 4 Species accumulation curves for both structured (blue triangles) and unstructured (gold circles) sampling methods.

Shaded areas represent the standard error. Data were obtained as described in Fig. 2.

Shannon diversity was generally quite variable in the unstructured sampling between counties and months, while the diversity observed by structured sampling varied less over space and time (Fig. 5). For diversity, model selection only slightly favored the inclusion of an interaction term (AIC = 89.5 without interaction, = 87.3 with interaction).

Figure 5 Boxplot comparing observed species diversity of odonates across five focal counties in the northeastern United States for structured and unstructured survey methods.

Data were obtained as described in Fig. 2.

Unstructured and structured sampling efforts captured strongly overlapping odonate species composition (ANOSIM, p = 0.09). In general, structured sampling captured a subset of the total community observed by unstructured sampling (Fig. 6). Permutational ANOVA found that community composition varied significantly by month (R2 = 0.29, p = 0.02), site (R2 = 0.22, p = 0.01), and sampling method (R2 = 0.07, p = 0.01).

Figure 6 Non-metric multi-dimensional scaling of communities of odonates observed through for both structured (blue triangles) and unstructured (gold circles) sampling methods.

Red crosses indicate centroids for the frequency of a given species. Unstructured samples are extracted from records contributed to the Global Biodiversity Information Facility database from iNaturalist and Odonata Central community science contributions. Structured surveys were completed by a trained individual conducting timed observations.

Discussion

For the focal counties and time periods, we found that unstructured community science and structured ‘expert’ sampling captured similar, but not identical patterns of Odonate biodiversity. For models for all biodiversity parameters performed better with an interaction effect between sampling method and sampling, suggesting that there are different patterns in reporting bias between sites per method. In general, structured sampling methods performed relatively consistently between sites while unstructured sampling methods had more variation, with each extreme end of the transect having more reported species richness than the intermediate data gap region. Given the dramatic differences in sampling structure between the two methods, any consistency in findings between unstructured and structured methods is likely a signal of a very strong underlying biological pattern (Boyd, Powney & Pescott, 2023). Our structured sampling approach was conducted over a limited but consistent time period, at specific sites selected for accessibility and perceived appropriateness of habitat, by consistent individuals following a known observation protocol. In contrast, our unstructured samples were compiled across larger geographical areas (entire counties, but without specific attention to selecting sites with appropriate habitat), over a longer time period, submitted to public databases with no consistent information recorded regarding sampling effort. While the unstructured samples produced approximately three times the number of individual observations across the transect, because records were so sparse in several sampled areas, this method could not demonstrate whether these sparse areas were due to low odonate density or low sampling density.

Despite the potential unreliability of the unstructured samples in providing estimates of relative population size, we also found that unstructured, community science sampling uncovered a much greater overall richness in Odonates than the structured sampling method (Fig. 4). Not only did unstructured sampling simply find more species, resampling analysis suggests that structured sampling curves saturate at lower numbers of individuals sampled: essentially, more structured sampling effort using our design is unlikely to uncover more new species. This pattern may occur due to several factors: structured sampling may preferentially bais data towards particular species with biologies most likely to be detected by the methodology. Furthermore, community scientists may be more likely to report novel observations, made opportunistically, whereas structured sampling would prohibit the reporting of incidental observations made outside of the experimental protocol.

Even with these differences, both sampling methods reported community composition similarly (Fig. 6), though structured sampling showed less variability, likely because of the targeted nature of this sampling in a single focal habitat, rather than representing an aggregation of records over space and time as the unstructured data, as well as a more consistent sampling effort. In the unstructured data set, reported species richness and abundance varied across the transect route, with extreme ends showing higher measurements than locations intermediate. In general, areas with higher biodiversity parameter values among the unstructured observations correspond to human population density, creating an opportunity for more sampling effort (Table 1). However, the notable low observation rate of odonates, regardless of sampling method, at several sites within the transect suggest that sampling biases due to effort do not entirely explain these trends. For instance, while we observed consistently high species richness in county C by structured sampling, where no odonates were recorded by unstructured samples, we observed the lowest richness by structured samples and the second-lowest richness by unstructured samples in county D (Fig. 3).

The observed data gap was most apparent in the unstructured data set, with structured efforts reporting a more consistent, but lower overall, diversity between sites. In contrast, unstructured efforts outperformed structured efforts in locations with more overall reports. This was particularly true for Ohio sites (A and B) compared to non-Ohio sites, likely as a result of the efforts of the Ohio Odonate Survey, whose state-wide efforts have contributed over 150,000 open-sourced records over the last two decades.

Our findings agreed with numerous studies supporting the viability and general reliability of community science in conservation, although our findings are consistent with others that unstructured methods are best used for species detection rather than estimates of population size (Lauret et al., 2021; McKinley et al., 2017) and Odonata biodiversity research (Patten et al., 2019; Rapacciuolo et al., 2017). When community science results follow the same general trends as structured sampling, findings can be reliably incorporated into occupancy models (Lauret et al., 2021) and support interpretation of findings when scientist-collected data are sparse (Walker et al., 2016). However, while inconclusive and of limited spatiotemporal scale, our observations of the data gap region mirror concerns about sampling biases and decision-making in community science that have also been the focus of many other studies (Archer et al., 2014; Bowler et al., 2022; Johnston et al., 2020; Millar, Hazell & Melles, 2019; Ruete, 2015). Our study held a similar constraint to many community science programs: sites were selected based on accessibility so that we were able to get to them in a short sampling period. This site selection created a bias towards habitats near transportation routes, most likely to be frequented by other humans. Data from community science programs (and structured surveys alike) are more likely to be complete in areas where more people are likely to go (Millar, Hazell & Melles, 2019). In one study, areas of public concern were suggested to have been oversampled by community participants (Jollymore et al., 2017). Similarly, other studies have criticized lay-user-generated geographical data, questioning their reliability, quality, and overall value (Flanagin & Metzger, 2008). In one study, areas of public concern were suggested to have been oversampled by community participants (Jollymore et al., 2017). Similarly, other studies have criticized lay-user-generated geographical data, questioning their reliability, quality, and overall value (Flanagin & Metzger, 2008). However, these issues may not be easily addressed, as an increasing focus on data quality could come at the cost of widespread accessibility (Parsons, Lukyanenko & Wiersma, 2011). However, when data are sparse, information about species assemblages and extents are often skewed (Johnston et al., 2020). A possible antidote to these biases that still capitalize on the energy and extent of community science surveys is to provide a semi-structured survey method that utilizes species lists and recording sampling effort, essentially providing a ‘denominator’ to estimate not just what and how many were observed, but over what spatial and temporal extents were searched. Using recurring and semi-structured methods as an addendum to bioblitz-style surveys provided improved estimates of population-sizes for birds and insects, however these methods were more labor intensive and required additional training for community science participants (Gigliotti, Franzem & Ferguson, 2023).

While community science records covered several years of flight seasons, field truthing efforts were only able to cover the peak season of 2019, as travel restrictions arising from the COVID-19 pandemic prevented repeated structured sampling in the following years. Once-per-month structured sampling allowed us to cover a much larger distance but introduced a risk of underreporting highly migratory species, like the green darner (Anax junius), whose swarms tend to attract large numbers of community science reports nationwide. While this spatio-temporal snapshot is not broad enough to speak for the applications of community science as a whole, it has provided an interesting case study for an understudied region, upon which future studies can be built.

Lastly, we acknowledge that community science data is limited by the common usage patterns of their platforms. In particular, it is likely that the abundance reported by unstructured community science platforms like iNaturalist is dramatically skewed from actual population size, as many community scientists report based on species presence-absence, instead of reporting the total number of individuals seen. In general, unstructured biodiversity data can more reliably document presence of a species, and sometimes, indirectly, absence in well sampled places (Guzman et al., 2021). Estimates of abundance from unstructured data are less reliable, but in some cases, relative population size estimates may be inferred in certain data-rich scenarios (Perry et al., 2022).

Conclusions

These observed differences in biodiversity patterns serve as an important case study, highlighting the productivity and broad geographical reach of large, long-term, community science efforts. While the underlying causes of this data gap region remain a subject for future studies, the absence of such in the structured sampling alludes to a human-driven source of bias in community sampling efforts.

Although community science has been shown to be capable of generating large amounts of observations, the actual efficacy of community science-based reporting for this taxon appears to rely heavily on external factors pertaining to how people interact with nature. We expect that population density and accessibility may be a large predictive factor of community engagement, underscoring a need for further research into engagement patterns in community science efforts and potential biases that may arise from them. Ultimately, for studies interested in range and biodiversity, community science data could represent a thorough, crowd-sourced alternative to traditional data sources, especially in areas with prolific community initiatives.

Future efforts will be aimed at identifying and analyzing other sources of inaccuracies or biases within community science efforts. More particularly, future studies will focus on the effects of coloration and visibility of common odonates on community science reporting rates, as well as evaluating how community science efforts compare to historical museum collections for this region.

Additional Information and Declarations

Competing Interests

Author Contributions

Data Availability

The authors declare there are no competing interests.

Christian M. Bullion conceived and designed the experiments, performed the experiments, analyzed the data, prepared figures and/or tables, authored or reviewed drafts of the article, and approved the final draft.

Christie A. Bahlai conceived and designed the experiments, analyzed the data, prepared figures and/or tables, authored or reviewed drafts of the article, and approved the final draft.

The following information was supplied regarding data availability:

The data and code are available at Zenodo: Bahlai, C., & Bullion, C. (2024). Odonata Gap Code. Zenodo. https://doi.org/10.5281/zenodo.12571071.

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
