# Peer review of "Data gap or biodiversity gap? Evaluating apparent spatial biases in community science observations of Odonata in the east-central United States"

_PeerJ, doi:10.7717/peerj.18115_

## Round 0.1 · original submission · Major Revisions

First off, thank you to the reviewers for their detailed and comprehensive reviews. I appreciate the time and effort that you took to apply your expertise to assessing this manuscript.

While the reviewers both find this study interesting and valuable (and I agree), both reviewers also suggest major revisions. The suggestions are pretty diverse and include tightening up language, ensuring things like methods or results are in the correct sections, a variety of excellent statistical recommendations, etc. One reviewer also mentioned the need for metadata in the data sets (thanks for including all of the data and code, as per journal policy).

I believe that these suggestions are all doable and will really further improve this manuscript and increase its value. I am looking forward to reading the revised version and your detailed response/rebuttal.

As this is a major revision, the revised manuscript will need to go for a second round of review, hopefully from one or both of the initial reviewers.

·

Excellent Review

This review has been rated excellent by staff (in the top 15% of reviews)
EDITOR COMMENT
Thank you for your on-time and exceptionally detailed review. I appreciate that you discussed aspects of wording (e.g. citizen vs. community science) and got into details about experimental design.

Basic reporting

I really enjoyed reading and reviewing this article. It's a well-prepared, "clean" story. The code is well annotated and the raw data is shared. I love the premise of the article, and think that these kinds of comparisons amongst methods are critical - particularly when we have access to lots of biodiversity data that might be used for making better-informed conservation decisions.

This paper would benefit from paring down the introduction a little bit, and further developing the discussion.This is a little off-balance, and I think you could make a bigger impact with your story if you took a little more time to explore some of the nuance of the story. I would think quite a lot of the material from the introduction might be more effectively "digested" by the reader once they've had the time to read through the methodology and results. There are a lot of neat things hinted about in the discussion, that don't get their time to shine. I would advocate you expand on some of these intriguing points, and make your story even more interesting to the reader.

There are some synonyms bouncing around in here in equal use, that caused a bit of confusion. I would recommend standardizing for clarity purposes. These include:
a) Citizen vs. Community Science
b) Field survey vs. systematic survey vs. structured sampling vs. sampling vs. transect samples vs. ground truthing vs. field truthing? Are these all the same thing? I think usually they are. Please pick one of them, and use it consistently throughout.
c) I quite like your use of structured and unstructured. I would advocate explaining each of these, and using them throughout.

Here are some places where I thought terminology could have been explained by alternate means to improve clarity:
L60: From iNaturalist, or something else, some combination
69: "user-generated geographical data"
81: "low observation rates" - of rare habitats, or rare species? Please clarify
119: Dual use of "study" in 119 made this a little confusing to follow.
154: Do you mean the same person was the observer, and the same the recorder at all sites, during all times?
181: What do you mean by "calculating Shannon diversity for each metric".
181: This is reading like Species Richness and Shannon's Diversity are the same thing. Please give this section a closer read over for clarity.
181: By "Source" - do you mean "Origin"... as in, "community science" OR "ground-truth survey"
188: What do you mean by "species richness counts"
190: Should this maybe be "In contrast" rather than "Likewise". Though I really like the way that you broke down how you would be interpreting these models. Nice one!
198: Can you explain why you did the ANOSIM and HoV? It's kind of the only thing you haven't justified in the methodology.
205: C. maculata
212: They're not really the same timeframe though are they? Because one was all from 2019, while the other was three years combined?
219: The term "reporting behaviour" is introduced here, and it is confusing. I would make sure this sentence is extra sharp so your reader does not stumble over as it, as it's an extra important sentence.
244: Not clear what this sentence is getting at - can you please rephrase?
249: "Less overall" -> "Lower"

Rather than the long list of citations in 260, I think it would be helpful to dig a little more into the meat of what a handful of these studies had to say.

Break down the sentence beginning on L109 - this is quite long and hard to follow.

I think it's always helpful to remind the reader (in a sentence) that a Lower AIC is better (to help the forgetful people out there)! ^_^

Comments on Figures
Figure 1. Scale bar missing. North arrow missing. Political boundaries over the heat map would be useful for those w/o a great understanding of central US political boundaries.

Figure 2. No need for a legend. Everything covered by x-axis.

Figure 3. Would swap the order of the categories here or elsewhere so it's consistent. The labels could probably go on the X axis, so they could be read without turning ones head.

Figure 4. Use the x-axis to label your sites. Remind the reader in the caption how they're arranged N-S. I think N-S might be a more logical way to present them actually.

Figure 5. Can you indicate the grey area - 95% CI or something else? Why only 4 sites for the unstructured data?

Figure 6: No need for a legend. Everything covered by x-axis.

Experimental design

The study looks to compare the diversity of odonates across 5 counties in the Central US. Two of these counties are "data gaps" where there is very little data available about odonates.

Clarifications:
I've got a couple of questions about design. When you visited each of your study sites for the ground truthing sites is it: One Lake per county, each lake sampled during 3 different months (Jun, Jul, Aug), from 3 different "vantage points".

- How spatially separated were the vantage points?
- How far into the lake did the observer stare? Use of binoculars?

Here are some of my concerns:
1. If you're using a single lake as an "area" from the 2019 season, why is this being compared to the county as a whole? Wouldn't a more meaningful comparison be community science data that is from that same Lake, rather than the County more broadly? Can you please explain why this comparison is the one that you made.

2. I don't think I fully grasp the details of understand what the "Observation Units" are from the community science data based on your description. Can you think of a way to explain this a little more clearly? Maybe a flow chart could help the reader out.

3. This idea of abundance is a little bit odd to me, as it seems (just based on what we know about sampling effort, and the limitations of citizen science data), that the abundances would differ inherently. Can you talk about this a little more? Lastly, Why is abundance being measured on a per species basis? [L225]. I think the inclusion of abundance data needs to be justified a little more.


I also think there needs to be more attention in describing how the different community science datasets differ from one another. I'd also like to see what the contribution of these different data sources are to the unstructured dataset. Are there any protocols in place? Who validates these? Can anyone dump this data into a respository, are there quality control checks in place. This might be a useful thing to share in tabular or paragraph form in the methodology section

Validity of the findings

I disagree with the conclusion that you made in L240, at least how it's written. Take for example Figure 2, Site A: How is this measure of biodiversity (here reported as richness) comparable? Maybe this makes more sense w/ the Shannon's Index measure, but those indices are difficult to intuit.

You mentioned that structured sampling showed less variability. This is because you sampled a much smaller number of habitats. Isn't this is what we would expect? Could you explain that a little more please?

You talked about the lack of unstructured data probably being due to anthropogenic factors. Could you maybe do a quick "tally" of *all* iNaturalist data (# of observations), and number of users, for each of the Counties in the year of the survey, and maybe population density and size of these counties? These could be added into Table 1, and could better contextualize the trends that you're seeing as being about community science or engagement w/ nature, rather than being more narrowly about dragonflies/damselflies. This might help this paper find a larger audience.

I am having trouble following your line of reasoning in Paragraph beginning on 255. What are the concerns that your study is mirroring? You've recorded Odonate data from one site in the county, and you're comparing that to a much larger spatial area. Can you walk that out a little more please?

To your last point L272, for sure! This is why I think the abundance data might have less value. Again, I'd really like you to explain why this was a variable that you considered so heavily in this study.

Finally, you haven't really dug into the biology/ecology of these species. What kinds of species (or what specific traits) make a species more likely to be included within your structured survey. Are you missing the forest dwelling species? The stream-loving species? The ones that w/ very brief adult lifespans? This would be good to contextualize (even very briefly). Maybe a Venn-diagram or the like, or show some of these NDMS analysis that were discussed in the methods, but to the best of my knowledge were not remarked on any further. It would be good to dig a little more into what traits made a species less likely to turn-up in your survey, and then talk a little more about the limitation of using a single lake compared to county level data.

Reviewer 2 ·

Excellent Review

This review has been rated excellent by staff (in the top 15% of reviews)
EDITOR COMMENT
Thank you so much for applying your expertise to reviewing this manuscript. Your review was very detailed, and I particularly appreciate your attention to and suggestions about the statistical methods.

Basic reporting

The manuscript was clear and professional with sufficient background and context.
Raw data and code is shared, but needs metadata.

I thought the introduction was very clear about the background to the study, why odonates are a good organism for these comparisons, and the objectives of the study. My comments are mostly minor.

Experimental design

The methods section was also clear, however some methods that are in results, are not mentioned in the methods (species accumulation, anosim, etc.). Why was month included in the model if you are compressing species richness by other temporal and spatial factors?

I also suggest that the authors include rarefaction curves and compare predicted species richness, given sampling effort (# of individuals), as that is more robust than raw species richness.

Validity of the findings

I’d like to see more of the results and effect sizes included in the results section. For your GLM, please include the models results in a table. This could also include pairwise comparisons of the sites by sampling type to clearly show differences are similar (although in your code, it looks like there is an interactive effect, suggesting sampling method influences inferences when comparing sites). Likewise for the AIC and community composition results. That will help make your results clear for the reader.

It is not completely clear to me how the authors arrive at their conclusions from the results presented. Perhaps more interpretation of the commmunity composition results (with figures) will bolster their conclusions.

Additional comments

The authors conducted a study comparing unstructured community science observations with structured surveys along a gradient of counties that vary in population density. They tested the hypothesis that dragonfly observations are underreported in low-density rural areas. I commend the authors for a clear methodological study evaluating the use of inaturalist data for insect community ecology. I was very interested in reading what they had to find, and I think similar studies like this will be very valuable for future ecological studies using these data. However, I found the analyses to be a bit confusing. There were several places where I noted that analytical approaches could be improved to better test their questions, as well as improvements to the figures. I hope these suggestions are helpful in improving this manuscript. The authors supply the data and code, however, the datafiles need metadata included with them.

Introduction:
Line 62: I think this paragraph could benefit from describing that other biases may exist in community science data in addition to spatial biases, e.g., temporal, and taxonomic (size, color, behavior, traits), etc.
Line 102: “While this region neighbors the 102 National Radio Quiet Zone (NRQZ),” It wasn’t clear to me why this line was included. Can you clarify?
Line 116“Thus, we predict that between site community trends will differ by human population density in the citizen science data but will be more equally distributed across sites in controlled expert surveys.” Is this really what you aimed to test? Later, your modeling framework suggests that (assessing the interaction term). But your figures all show paired comparisons. I think both are valid, but those are two approaches to the question. 1) Are relative comparisons valid? 2) Is species richness estimates valid?
Also, something that jumped at me is that the Appalachian is a very different ecosystem too. But given you have paired methods (comparing community sci to ground truthing), and are interested in testing relative differences, then that is ok. Perhaps remind the reader that that is the core comparison you are interested in.

Methods:
Line 135: chosen to be approximately equally spaced, with bodies of water in naturalized areas that were reasonably accessible from roads or campgrounds
This reads like you intentionally geographically biased your sampling as well? Why not have stratified random locations? It’s not clear to me why you want to target these easily accessible locations. In this sampling framework, it reads to me like you are testing the methods of sampling (via transect or opportunistic) rather than geographic bias (at least at smaller spatial scales).

Line 156: Community science data was represented by a combined Global Biodiversity Information
Facility dataset, containing reports originating from iNaturalist and Odonata Central (GBIF.org,
158 2021).”
If from GBIF, were museum specimens, etc. filtered out? I assume so, but perhaps state directly.

Line 162: compatible with community analysis, “Can you be specific here? These high zeros makes it incompatible for temporally explicit community analysis, but not community analysis in general.
Line 168: Cite these after their respective packages, not in abc order.

Line 181, Why use Shannon diversity and species richness and not a more robust method to preduct species given sampling? Species rarefaction curves, setc. Species richness is always imperfectly detected, no matter the method. How we account for that would better assess the reliability of community science data

Line 196: “To examine differences in community composition among sites, sampling periods, and sampling 196 approaches, ” Can you expand on how you compared community composition while controlling for site? Or are all these conducted separately? A permanova could be one way to test the influence of sampling approaches on community composition while controlling for site. I also don’t see much results, figures or interpretation of this later. But this was a part of the analysis I was really interested in.

RESULTS
Line 203: “This study included 1,573 observations, with 381 originating from structured sampling efforts 203 and 1,192 originating from community science efforts.” These are very different sampling effort. It would be interesting to see what the predicted richness is controlling for samples using rarefaction curves.

Line 215: Oh wait, there was species accumulation curves . at least with increasing site sampling This should be in the methods.

Line 233: “Likewise, beta dispersion analysis “This needs to be described above too. How are these two tests addressing your question? Also, where are the community composition figures? It feels like this was completed, but not really used to it’s full potential. Why is there no interpretation of this analysis?

Line 235: I think you mean Analysis of variance? What is this testing? An ANOVA on community variance?

Discussion:
Line 240: For the focal counties and time periods, we found that unstructured community science and structured sampling performed comparably well in reporting Odonate biodiversity. But your interaction had an effect, meaning, that method impacted comparison of sites. Am I missing something? I would interpret that as meaning that the methods matter. It’s not clear to me how you are measuring ‘consistency’ throughout.

Line 273: “In particular, it is likely that the abundance reported by community science is much higher than actual values, as many community scientists report based on species presence absence as a whole, instead of reporting the total number of individuals seen” This is always true for inat. Is it not for the Ohio Odonate surveys?

Line 287: I think you should be transparent that this study is only looking at 5 discrete regions and species richness/abundance and composition. Whereas, inat data is/can be used to predict probabilities over much larger areas. Using inat data in that way may still bias species distribution predictions toward areas of high population densities, regardless if estimations of species richness is reasonable. So while I agree that there are other biases to consider (that perhaps could have been looked at with this dataset?) I think there is more we should be considering for spatial biases as well.

Figures

Figure 1: minor detail but throughout the i in iNaturalist should be lower case.
Figure 2: It would be great to show the raw richness comparison, alongside the predicted species comparison. It could be that it really doesn’t matter when you account for sampling effort. Here, the biases are obviously related to how many observations.
Figure 3 and 4 should be combined to show the different in each location by sampling type.
Figure 5. Rather than have a species accumulation curve. The authors should have a rarefaction estimate of species diversity that accounts for individuals sampling and compares the two methods. Showing species accumulation by the increase in sites isn’t as meaningful to addressing the question. This figure is just showing that you have higher species observed as you sample more sites, which is not surprising or related to the questions as stated.
Figure 6 This could be combined with one of the earlier richness figures to free of space for an ordination plot of something to show community differences.

---

## Round 0.2 · Minor Revisions

One of the previous reviewers has re-reviewed this manuscript and has recommended minor revisions. The nature of the revisions indicates that they are, indeed, minor -- so I suspect we are close to being able to accept this for publication in PeerJ. Your revisions will not need to go out for another round of review, I will make a decision at that point.

Please note that I am away on family vacation from tomorrow until Friday 16 August. So if you return the manuscript sooner than that, I won't be available to render a final decision until Monday 19 August at the earliest (probably a bit longer than that considering what my inbox will likely look like upon my return).

Thanks to the reviewers throughout this process, and thanks to the co-authors for their work on the manuscript.

·

Basic reporting

The manuscript is looking really good. I appreciate the thoughtful and clear responses to the previous submission. The data is clearly reported, and the authors have done a really good job explaining the methodology - with particular attention to explaining the models (which is so appreciated).

I think it would be helpful to remind the reader about your sampling effort in the captions in Figure 2 & 3. Especially when you discuss the idea of County C not having a single observation of an Odonate. I think it would also be helpful to explain sample size within these figure captions, and describe what constitutes a sample.

121-125: Please always use the conventions of structured and unstructured sampling.

I still think the introduction is way too long. You’re potentially losing the reader, on what I think is a really good contribution.

Simplify: Intro paragraph 1+2 (aim to reduce the words by ½)
Combine and condense paragraphs beginning on: 67+78). You can reuse this writing within the discussion (I think it belongs there very nicely).
Combine and condense paragraphs beginning on L90+100.

Experimental design

The experimental design is very clearly described now. I think the major limitation of this approach is needs a little more attention in the discussion. The fact remains is that these comparisons (structured vs. unstructured) are really not so comparable, because they are sampling different things (many streams, ponds, and lakes within a county) to (single accessible lakes near roadways). It is not surprising that the number of species is different between these samples.

Please expand on this in the discussion in a little more detail. Maybe there is another similar study that has made a comparison, and this could be contrasted/compared against your work?

Validity of the findings

Please double-check the Figure 5, sub-panel B&E: What's going on there with your Shannon Diversity?

Table 1: Please explain that iNaturalist users - reflects the number of individuals reporting Odonate data within your sample (or is it total number of active users within the county?)
I would dig in a little more to the rarefaction curve. Your structured survey is going to become asymptotic at a much lower richness. This should be discussed in greater detail at L260 (this dovetails with comments above in section 2):
- Does this reflect a limitation of your structured samples
- Does this reflect less suitable habitat for odonates?
- Or is this trend being driven by beta-diversity ?
- Or does your sampling approach not suited to detecting certain species?

281: I am not convinced that your data is telling us anything about the reliability of community science data. This is not specifically addressed. Please re-word, and make a stronger argument – otherwise please remove this statement. Just does not seem congruent to me with your results. The rest of the paragraph is great!

Additional comments

Needs clarification:

41-42, 42-43,79-80: The context making this study relevant to yours is not clear - more information needed to understand the connection.

106-107: The point about the NRQZ needs clarification please. Not following your logic here.

117: What do you mean by "These regions"? Please be specific.

263: Do you mean community composition?

271-272: Can you refer back to a Figure or Table to strengthen your argument about low observation rates. Talk about these counties specifically.
277: Please clarify that the Ohio sites are A+B (Earlier, you mention referring to the sites by these names – but then tend to flip back and forth). Please be really consistent

311: These is not the case for eBird. I think it depends very strongly on what the data source is. I would consider making this claim a little less sweeping for accuracy?

Throughout the manuscript - when mentioning American states, please specify that these are American states. This is not necessarily going to be common knowledge for a global audience. Alternatively, you can refer to these areas less specifically as regions or by habitat.

References need work:
Duplication of Rapacciuolo (2016)
Some italicization of species names missing (e.g. May et al. 2017)
Challenges with capitalization consistency (e.g. Wickham 2011, Zarnetske 2012)
Typo on Figure 2 Caption: iNaturalist. Same typo on Figure 6.
Typo 226: uniform
255: Missing “of”
Table 1: Include size of the county in km^2
Table 1: Is population per mile squared conventional for reporting? If no, please convert to km2.

---

## Round 0.3 · accepted · Accept

The authors have addressed all of the final, minor comments. This MS is now ready for publication in PeerJ. Thanks to the reviewers for their time and expertise, and to the co-authors for the helpful and detailed responses.